# The Ubiquitin Proteasome System in Hematological Malignancies: New Insight into Its Functional Role and Therapeutic Options

**DOI:** 10.3390/cancers12071898

**Published:** 2020-07-14

**Authors:** Antonella Di Costanzo, Nunzio Del Gaudio, Lidio Conte, Lucia Altucci

**Affiliations:** 1Department of Precision Medicine, University of Campania "Luigi Vanvitelli", Vico L. De Crecchio 7, 80138 Naples, Italy; nunzio.delgaudio@unicampania.it (N.D.G.); lidio.conte@unicampania.it (L.C.); 2Center for Genomic Science of IIT@SEMM, Fondazione Istituto Italiano di Tecnologia (IIT), 20139 Milan, Italy

**Keywords:** UPS, hematological malignancies, PIs

## Abstract

The ubiquitin proteasome system (UPS) is the main cellular degradation machinery designed for controlling turnover of critical proteins involved in cancer pathogenesis, including hematological malignancies. UPS plays a functional role in regulating turnover of key proteins involved in cell cycle arrest, apoptosis and terminal differentiation. When deregulated, it leads to several disorders, including cancer. Several studies indicate that, in some subtypes of human hematological neoplasms such as multiple myeloma and Burkitt’s lymphoma, abnormalities in the UPS made it an attractive therapeutic target due to pro-cancer activity. In this review, we discuss the aberrant role of UPS evaluating its impact in hematological malignancies. Finally, we also review the most promising therapeutic approaches to target UPS as powerful strategies to improve treatment of blood cancers.

## 1. Introduction

Post-translational modifications are an intriguing mechanism through which cells regulate protein homeostasis. Protein ubiquitination is the major post-translational modification that is responsible for either protein biological function alteration or specific protein degradation by the 26S proteasome [1]. A multitude of proteins controlling several aspects of cell physiology, including cell cycle progression, differentiation and apoptosis, undergo ubiquitination and degradation processes mediated by the ubiquitin proteasome system (UPS). In addition, several studies have demonstrated alteration of UPS in cancer cells along with a higher rate of protein turnover compared to normal cells. In this scenario, it became clear that UPS plays a fundamental role in the regulation of a wide range of biological process and that deregulations in the UPS activity are responsible for different pathologies, including hematological malignancies [2]. Consequently, targeting UPS effectors has become an alternative strategy for cancer treatment. A great number of proteins with a functional role in the onset and maintenance of hematological cancers are regulated by ubiquitin modifications; thus, alterations in the ubiquitination and proteasome-mediated degradation pathway have been reported to play a critical role in hematological malignancies [3,4,5].

Several studies have described the higher activity of proteasome and its aberrant role in hematological malignancies making leukemic cells particularly sensitive to proteasome inhibitors (PIs). The approval of bortezomib (Velcade) (BTZ), the first best-in-class PI for the treatment of multiple myeloma (MM), clearly opened up new perspective for the treatment of other malignancies such as mantle cell lymphoma (MCL), chronic lymphocytic leukemia (CLL) and acute myeloid leukemia (AML). Therefore, research and development of new PIs, as well as the studies on PIs treatments combined with other agents used in the therapeutic setting of leukemia and lymphoma patients, represent an active and promising research area.

In this review, we focus on how deregulation of the UPS components contributes to tumorigenesis and discuss the status quo of the different therapeutic molecules targeting UPS in hematological malignancies.

## 2. The UPS Machinery in Cell Biology

The 26S proteasome is a macromolecular complex consisting of a catalytic 20S and regulatory 19S subunits responsible for the degradation of about 80% of the targeted intracellular proteins by an elegantly controlled mechanism. Cellular proteins undergo ubiquitin proteasome-mediated degradation fundamentally through two sequential steps. First, proteins are tagged with monomers or polymers of the ubiquitin chain; next, they are degraded by the 20S proteasome subunit [6]. Ubiquitination is the first step in the UPS. Specifically, proteins targeted for degradation are tagged by ubiquitin, a small peptide of 76 amino acid residues, through coordinated actions of serial enzymes including the ubiquitin-activating enzymes E1 (Ub-activating enzyme ~2), ubiquitin-conjugating enzymes E2 (Ub-conjugating enzymes ~38), and the ubiquitin E3 ligases (Ub ligase enzyme ~700) that recognize specific motifs and catalyze the formation of polyubiquitin chain on the substrate proteins. Due to their critical role in the selective binding of protein substrates, E3 enzymes are distinguished in multi-subunit families that confer a high degree of substrate specificity. Two main different subclasses of E3 enzymes, based on their structural and biochemical characteristics, have been characterized: HECT (Homologous to the E6-Associated Protein C-Terminus) and RING (Really Interesting New Gene).

RING E3 ligases contain a zinc-binding domain named RING (Really Interesting New Gene) or a U-box domain that does not contain zinc. The RING and U-box domains bind the ubiquitin-charged E2 and promote ubiquitin (addition) transfer [7]. HECT E3 ligases catalyze ubiquitination of the substrate protein by a two-step reaction: ubiquitin is transferred with a trans-thiolation reaction to a catalytic cysteine on the E3 and then from the E3 to the target substrate [8].

### 2.1. Mono and Multi-Polyubiquitination

Protein monoubiquitination or multi-ubiquitination depend, by the attachment of a single ubiquitin molecule, on single or multiple lysine (K) residues, respectively. While the functional role of polyubiquitination into proteins clearance by proteasome-mediated degradation has been largely recognized, monoubiquitination is primarily implicated in the regulation of some intrinsic biological aspects of target proteins, such as DNA damage repair, chromatin regulation, intracellular protein localization, trafficking and regulation of protein complex formation [9,10,11,12]. For example, H-Ras ubiquitination are responsible for its relocation from the plasma membrane to endosomal sites; conversely, K-Ras monoubiquitination modulates its affinity for downstream effectors such as Raf-1 and phosphoinositide 3-kinase (PI3K) [13]. Additionally, monoubiquitination of actin-binding protein filamin B regulates the trafficking nucleus–cytosol and cytoplasmic localization of HDAC7 in response to VEGF signaling [14]. For extensive discussion about the functional role of monoubiquitination, refer to Nakagawa et al. [15]. Recent studies have correlated monoubiquitination to proteasome-mediated degradation by 26S [16]. Intriguingly, it seems that the size of substrates plays a pivotal role in the affinity of the Ub-modified substrate to the proteasome. In particular, it has been reported that substrates smaller than 150 amino acids are degraded following monoubiquitination, while longer substrates require longer Ub-chains.

### 2.2. Polyubiquitination

The polyubiquitination process take place from the binding of ubiquitin polymers to either single or multiple K residues of target proteins. Several “types” of polyubiquitination have been identified due to the different binding of ubiquitin moieties at specific K residues (K6, K11, K27, K29, K33, K48 and K63) or the N-terminal methionine residue (M1) [17]. It has been found that each of the polyubiquitination linkages has a distinct biological function. The most abundant Ub chain types are K48-linked chains, which predominantly mediate proteasomal degradation of target proteins, and K63-linked chains, which is responsible for a non-degradative role like intracellular trafficking, kinase signaling, and DNA damage response, along with other processes [18]. Additionally, “atypical” ubiquitin modifications (K6, K11, K27, K29 or K33) mediates a plethora of different biological functions. For example, K6-linked chains are involved in mitophagy regulation [19] and the DNA damage repair process [20], while K11-linked chains regulate cell division, transcription regulation and endoplasmic reticulum-associated degradation (ERAD) [21]. K27-linked chains function like recruiting proteins for DNA repair and immune response, whereas K29-linked chains participate in the ubiquitin-fusion degradation (UFD) pathway, mediating degradation of proteins, which is normally not targeted to the proteasome [22]. K33-linked chains are involved in Toll receptor-mediated signaling pathways [23].

Ubiquitination is a reversible process. A specific class of enzymes, the deubiquitinating enzymes (DUBs), is responsible for the removal of ubiquitin from modified substrates rescuing proteins from degradation and finally enhancing protein stability. At least 100 DUB genes are encoded by the human genome. They are classified predominantly in two classes: cysteine proteases and metalloproteases. DUBs are involved in key biological pathways regulating cell cycle control, apoptosis, maintenance of stemness and DNA repair. Accordingly, their deregulation has a functional role in oncogenesis. For an extended review on the functions of the several UPS components, refer to [24,25].

## 3. Deregulation of UPS in Hematological Malignancies

Dysregulation of each component of UPS (E1, E2, E3, DUBs and proteasome) can contribute to cancer pathogenesis. In particular, a certain number of E3 ubiquitin ligases are mutated, overexpressed or deleted in hematologic malignancies, resulting in the aberrant accumulation or degradation of their targets. Thus, the altered physiological proteolysis of several oncosuppressors and/or oncogenes leads to the alteration of hematopoietic cell growth, differentiation and survival.

### 3.1. Aberrant Role of E1 and E2 Ubiquitin Ligases

To date, UBA1 and UBA6 are the only 2 ubiquitin E1 enzymes identified, although several studies reported UBA1 as the predominantly E1 isoform involved in the ubiquitin proteasome degradation pathway. In contrast to other UPS-associated enzymes, E1 expression does not appear to be deregulated among primary leukemic and healthy cells. However, since it has been demonstrated that primary leukemic cells show an aberrant activity of UPS [26], the rationale for pharmacological inhibition of E1 enzymes derives from their extra-effective use in neoplastic cells. Genetic and chemical inhibition studies demonstrated that the UBA1 enzyme is involved in the accumulation of ubiquitinated proteins in leukemia and myeloma cells, and its inhibition is responsible for leukemic cell death by endoplasmic reticulum (ER) stress-mediated and unfolded protein response mechanisms [27]. Moreover, since the inhibition of E1 enzyme impacts cell survival similarly to proteasome inhibition, there is a great expectation on the development of specific inhibitors for their potential use to overcome some forms of drug resistance resulting from PIs.

The E2 family comprises approximately 40 members. Some of these have emerged as tumor modulators, regulating cell cycle progression and activating different oncogenic pathways. Several E2 enzymes are deregulated in cancer and some of these have been taken into account as biomarkers for diagnosis and prognosis. Ubiquitin-conjugating enzyme E2 N (UBE2N) has been identified as a novel therapeutic target in myelodysplastic syndrome (MSD) and AML. It is an ubiquitin-conjugating E2 enzyme that is implicated in the addition of ubiquitin chains on lysine 63 (K63) of TRAF6 and other substrates. Targeting UBE2N results in an impairment of cell viability, repression of innate immune signaling and induction of mitotic alteration in MDS/AML cell lines and patient samples [28]. Moreover, UBE2N, complexed with its cofactors Uev1A (UBE2V1) or Mms2 (UBEV2), is indirectly involved in the activation of NF-kB or DNA double-strand break repair [29,30].

Ubiquitin-conjugating enzyme E2 C (UBE2C) is overexpressed in several cancers such as breast, pancreas, colon, prostate, lymphoma and ovarian carcinomas [31]. In particular, UBE2C expression correlates with lymphoma aggressiveness [32]. Similarly, the highest UBE2C expression at both the mRNA and protein level are reported in Burkitt’s lymphoma.

In leukemia, bone marrow (BM) cells from acute lymphoblastic leukemia (ALL) patients showed upregulation of E2 ubiquitin-conjugating enzyme Q2 (UBE2Q2) compared to the normal counterparts [33] and ubiquitin-conjugating enzyme E2 A (UBE2A) mutations are specifically acquired during chronic myeloid leukemia (CML) progression, resulting in a lower UBE2A activity leading to an impaired myeloid differentiation [34].

### 3.2. Aberrant Role of E3 Ubiquitin Ligase

F-box and WD repeat domain-containing protein 7 (FBXW7) is an F-box protein containing seven tandem WD40 repeats belonging to Skp1-Cul1-F-box-protein (SCF) E3 ubiquitin ligase. It is involved in the ubiquitination and degradation of many oncogenes and transcription factors, including p53, c-Myc, c-Jun, HSF1, NF-kB and Notch. Accumulating evidence reveals that the abnormal expression of FBW7 is involved in the development of hematological cancers [35]. T-cell acute lymphoblastic leukemia (T-ALL) is the only hematological tumor that can be caused by the deletion of FBXW7 without the necessity of other cancer-promoting elements [36]. In adult T-cell leukemia/lymphoma (ATL), FBXW7 acts as a tumor suppressor protein; however, some mutations can turn it into an oncogene. FBXW7 mutants fail to degrade Notch intracellular domain degradation (NICD), causing an increased activation of Notch1 signaling that promote ATL cell proliferation and tumorigenesis [37]. Close and colleagues [38] found that FBXW7 mutations decrease Notch1-NICD binding and degradation, hence leading to its activation in CLL cells [38].

FBXL2 (F-Box and leucine rich repeat protein 2), an F-box protein member of the SCF E3 ligase family, displayed anti-tumor activity and its expression is suppressed in AML and ALL patient samples. FBXL2 mediates cell cycle arrest and apoptosis in leukemic and B-lymphoblastoid cell lines by ubiquitination and destabilization of cyclin D2, highlighting its antiproliferative effect in lymphoproliferative malignancies [39].

F-Box only protein 9 (FBXO9) is an important regulator of AML and its expression is low in the inv (16) subtype of AML patients compared to healthy bone marrow. Conditional knockout of FBXO9 in AML mouse model reveals that FBXO9 plays a functional role not only in leukemia initiation, but also in AML maintaining and disease progression. Moreover, loss of FBXO9 results in increased proteasome activity, making tumor cells more sensitive to bortezomib [40].

The proto-oncogenic Casitas B-lineage lymphoma (c-CBL) proteins are mutated in different categories of myeloid malignancies (approximately 10%) such as myelodysplastic syndrome, myelofibrosis, de novo AML and secondary acute myeloid leukemia (sAML), CML and ALL. Structurally, CBL proteins possess an N-terminal tyrosine kinase binding (TKB) domain and a catalytic C3HC4 RING finger (RF) domain that confer E3 ubiquitin ligase activity. They regulate receptor tyrosine kinase-mediated signaling pathways as well as Akt-PI3K and Ras-Raf-MAPK pathways [41]. Most of the mutations characterizing CBL cause loss of ubiquitin E3 ligase activity, resulting in impaired lysosomal or ubiquitin/proteasome-mediated degradation of tyrosine kinases, finally resulting in uncontrolled tyrosine kinase signaling.

S-phase kinase-associated protein 2 (SKP2) is another E3 ligase belonging to the Skp1-Cul1-Fbox (SCF) E3 ligase complex. It is a regulator of cyclins p27, p57 and p21 level, mediating their ubiquitination and proteasome-dependent degradation, thus exerting a key role in cell cycle control. Several studies demonstrated that SKP2 functions as an oncogene and is overexpressed in cancer including hematological neoplasms [42].

Specifically, SKP2 is overexpressed in diffuse large B-cell lymphoma (DLBCL) and cutaneous T-cell lymphomas (CTCLs) where it is associated with poor overall survival [43] and abnormal cell growth, respectively [44]. Overexpression of SKP2 in CML is due to BCR-ABL transcriptional activity correlated to PI3K/AKT/Sp1 pathway while its deregulated expression is drove by activated Notch signaling [45]. Myristoylated alanine-rich C-kinase substrate (MARCKS) overexpression and *CKS1B* activity seem to be directly correlated with upregulation of SKP2 and, consequently, p27Kip1 and p21Cip1 deregulation, finally resulting in enhanced MM growth and survival [46].

The WW domain-containing E3 ubiquitin protein ligase 1 (WWP1), involved in the maintenance of epithelial cancer, has recently been found overexpressed in primary AML patients and in AML cell lines, compared with hematopoietic cells of healthy donors. Particularly, WWP1 seems to be involved in the aberrant proliferation and survival of both primary AML blasts and leukemic cell lines through deregulation of p27Kip1 and autophagy signaling, respectively [47].

KLHL6 is a member of the kelch-like (KLHL) family of proteins. It has been found specifically expressed in lymphoid tissues where it is involved in B-lymphocyte antigen receptor signaling and germinal-center B-cell maturation [48,49]. Frequent mutations of the KLH6 gene have been observed for different B-cell malignancies such as diffuse large B-cell lymphoma (DLBCL), follicular lymphoma (FL) and CLL [50,51]. Choi et al. [52] revealed that KLHL6 negatively regulates NF-kB activation. They found that the cancer-associated KLHL6 mutants were unable to perform their ligase activity by blocking the interaction with cullin3, thus sustaining NF-kB signaling and promoting lymphoma formation [52].

The more exhaustive E3 ligases aberrant activity characterization and the extensive comprehension of how they finely control key substrates involved in cancer progression represent ambitious and hard challenges to the development of new therapeutic interventions in hematological neoplasms.

Taking into account the wide ranging of UPS regulators and the different impact they have on cancer pathways, it becomes clear how UPS represents a precious mine to be exploited for development of novel drugs useful for anticancer therapies.

### 3.3. Modulation of UPS by Virus Induced Cancer

DNA tumor viruses take advance of the UPS to regulate and control the host immune response. Viral immune evasion proteins are the instrument used by viruses to improve their survival and replication in infected cells. For example, Kaposi sarcoma herpesvirus (KSHV) has been found implicated as an etiologic agent of different diseases, including primary effusion lymphoma (PEL) and multicentric Castleman disease (MCD) [53]. KSHV has developed mechanisms to evade innate and adaptive antiviral immune responses; KSHV can manipulate the cellular UPS by encoding its own viral E3 proteins. Intriguingly, K3 and K5 are KSHV-encoded proteins that belong to a family of viral RING-CH-finger E3 ubiquitin ligases; they are able to downregulate the cellular surface exhibition of MHC-I proteins [54,55]. Furthermore, K3 and K5 are able to target γ-interferon receptor 1 (IFNGR1), causing its ubiquitination, endocytosis and degradation, thus bringing a downregulation of IFN-γR1 protein surface expression [56]. For detailed information on KSHV, refer to Goncalves et al. [57].

Additionally, the Epstein–Barr virus (EBV), a herpesvirus very common in the human population, has been found to dysregulate oncogenic proteins such as c-Myc by exploiting the UPS, leading to the expansion of EBV-infected cells, and thus resulting in the development of EBV-associated cancers [58]. For extensive information about EBV, it is possible to consult Masucci et al. [59].

## 4. DUBs Activity in Hematological Malignancies

Deregulation of DUBs is reported in solid cancer and hematological neoplasms in which the unbalance between ubiquitination and deubiquitination contributes to abnormal accumulation of key proteins involved in several biological pathways [60,61]. In particular, DUBs have emerged as promising druggable targets due to their accessibility to modulation by small molecule inhibitors.

Ubiquitin-specific ubiquitinase 1 (USP1) has been identified as a negative regulator of DNA repair mechanisms. USP1 catalyzes the specific deubiquitination of Fanconi anemia group D2 protein.

Fanconi anemia group D2 protein (FANDC2) is a key protein involved in DNA repair [62]. USP1 also deubiquitinates the monoubiquinated form of proliferating cell nuclear antigen (PCNA) [63]. Mechanistically, it exhibits a poor deubiquitinating capacity alone; thus, it requires the interaction with USP-associated factor 1 (UAF1) to improve its activity [64]. Inhibitor of DNA-binding-1 (ID1) protein, an activator of leukemic cell growth with high expression in AML samples [65], is another target of USP1. The latter deubiquitinates ID1, avoiding its proteasome degradation [66].

Ubiquitin-specific peptidase 7 (USP7) is a regulator of different cancer-related proteins, including p53, MDM2, FOXO4 and Tip60; consequently, USP7 is an interesting therapeutic target in cancer [67]. USP7 is overexpressed in CLL compared with normal donors and its inhibition induces cytotoxicity in CLL cells both in vitro and in vivo, impairing homologous recombination DNA repair through the E3 ligase RAD18 destabilization. Furthermore, USP7 inhibition promotes poly (ADP-ribose) polymerase 1 (PARP1) activation and nicotinamide adenine dinucleotide (NAD1) depletion, resulting in the activation of both apoptotic and necrotic cell death pathways [68]. Finally, USP7 inhibition synergized with chemotherapy drugs such as mitomycin C and cyclophosphamide in vitro and in vivo [68].

Ubiquitin-specific peptidase 10 (USP10) mediates the deubiquitination of several targets involved in cell proliferation, such as p53, Sorting nexin 3 (SNX3) and cystic fibrosis transmembrane conductance regulator (CFTR) [69,70]. Although USP10 functionally acts as a tumor suppressor in different types of cancer, it also exerts a proto-oncogenic activity in AML by regulating cancer stem cells properties [69,71]. Additionally, Liao and colleagues [72] found that USP10 can promote leukemic cell proliferation by deubiquitinating and stabilizing the S-phase kinase-associated protein 2 (SKP2) that acts as a co-regulator of BCR-ABL mediating its K63-linked ubiquitination and activation in CML cells [72].

Intriguingly, USP10 inhibition has been shown to promote oncoprotein FLT3 degradation, suggesting that USP10 could be considered as a potential pharmacological target for fighting FLT3- mutant AML disease [73].

Ubiquitin-specific peptidase 9, X-linked (USP9X) is an ubiquitin protease involved in the maintenance of DNA fork stability and in the regulation of DNA damage checkpoint response. It removes ubiquitin moiety from several substrates including afadin, adherens junction formation factor (AF-6), b-catenin, myeloid leukemia cell differentiation protein (MCL1) and SMAD family member 4 SMAD4 [74,75,76]. The controversial role of USP9X in leukemia has not been fully elucidated yet; some studies classify USP9X as an oncogene [74], while other studies reported it as an oncosuppressor, suggesting that it displays a context-specific function [77]. It has been shown that USP9X positively regulates the protein stability of induced myeloid leukemia cell differentiation protein (MCL1) [74,78], a BCR-ABL dependent target implicated in drug resistance and cell survival of hematopoietic diseases [78]. Two different studies reported the positive effect of USP9X targeting in hematological malignancies. In B-cell acute lymphoblastic leukemia, USP9X inhibition sensitizes neoplastic lymphoid cells to glucocorticoid (GC)-induced apoptosis [79]. Intriguingly, inhibition of USPX9 using G9 or P220477 induces apoptosis in FLT3-positive MV4-11 cells line and primary AML cells, suggesting a possible new therapeutic strategy against AML [80].

## 5. Targeting UPS

There has been growing interest in exploring components of the ubiquitin-proteasome machinery as possible therapeutic targets, and many efforts have been made on developing small molecules to modulate this pathway for therapeutic approaches. Based on the ability to inhibit or activate the ubiquitin pathway, compounds targeting UPS can function as inhibitors or activators (Figure 1). Here, we discuss recent advances made for targeting UPS, mainly focusing on hematological malignancies.

### 5.1. Inhibiting E1, E2 and E3 Ubiquitin Ligases

Several molecules targeting different E1, E2 and E3 enzymes have been synthesized so far and have shown promising anticancer activity; however, only a few of them have been evaluated in clinical trials (Table 1).

PYR-41 was the first permeable Uba1 (E1) inhibitor identified. PYR-41 attenuates cytokine-mediated nuclear factor-κB (NFKB1) and prevents the downstream ubiquitination and proteasomal degradation of inhibitor of nuclear factor kappa B subunit α (NFKBIA; IkB-Alpha). Furthermore, PYR-41 can inhibit p53 degradation, inducing its tumor suppressor transcriptional activity [81]. Although PYR-41 triggers apoptosis of malignant cell lines, including leukemic cells, and primary samples, the exact mechanism of action is still not completely understood [82,83].

MLN7243/TAK-243 is a highly selective compound against closely related E1 enzymes, although its efficacy is also due to the inhibition of the ubiquitin-like molecule NEED8 substrate addition. MLN7243 is considered the most promising E1 inhibitor and it is currently being evaluated in clinical trials with promising results [84]. MLN7243 treatment results in the accumulation of tumor suppressor proteins such as NRF2, p27 and IκB, inhibiting abnormal cell growth [85]. MLN7243 has showed anticancer properties against preclinical multiple myeloma models [86].

Inhibition of E2 enzymes results in a higher target specificity compared to E1, and thus provides a better selectivity and fewer side effects compared to an E1-targeting compound.

CC9651 was the first small-molecule allosteric inhibitor of ubiquitin conjugating enzyme 2 (UBE) R1 (UBE2R1), displaying the ability to inhibit, for the first time, the central step of the ubiquitination pattern [87]. CC0651 showed promising anticancer activity, although due to drug optimization problems, it never proceeded into clinical trials [88].

The small molecule NSC697923 was developed as inhibitor of UBE2N and it showed the ability to reduce proliferation as well as cell growth of large B-cell lymphoma cells [89].

However, despite the fact that many efforts have been made for developing additional E1 and E2 inhibitors, except for MLN7243, there are no molecules targeting E1 or E2 enzymes entered in clinical trials due to unresolved questions related to specificity or drug properties.

E3 is the largest family of enzymes and comprises distinct catalytic mechanisms. Targeting E3s translates into better selectivity and less toxicity compared to E1 and E2. Because of the high number of existing E3 inhibitors, we discuss the recent advances on the most promising inhibitors of selected E3 enzymes here. The E3 ubiquitin ligase SKP2 regulates cancer progression by targeting several tumor suppressors and many studies have documented its involvement in several cancers, including hematological malignancies [61,90]. Thereby, inhibiting SKP2 is a great potential for developing new anticancer drugs.

Two molecules, compound A (SMIP0004) and compound 25, have been reported to reduce SKP2 activity. Both compounds exhibited apoptosis induction and cell growth inhibition in MM cells as well as in other solid tumors [91,92]. Moreover, compound 25 has also shown synergistic effects in association with chemotherapeutic drugs [91]. Additional biological studies and structure modifications will further improve the potency of SKP2 inhibitors for eventually reaching clinical applications in cancer therapies.

By high-throughput screening, a family of cis-imidazoline derivatives, named nutlins, was identified as an inhibitor of MDM2. Nutlins directly bind to the MDM2-p53 binding pocket, inhibiting degradation of wild type p53, leading to cell cycle arrest and induction of apoptosis [93]. Nutlin derivatives entered in clinical trials for the treatment of leukemia and solid tumors [94]. Additionally, several small molecules have also been identified to disrupt MDM2-p53 interaction, hence affecting proliferation and survival of cancer cells expressing wild type p53. Among them, two MDM2 inhibitors, APG-115 and ALRN-6924, are currently being testing in clinical trials as promising anticancer drugs for hematological malignances [95,96]. Oridonin is a natural derivative compound exhibiting a broad range of biological effects, comprising anticancer and anti-inflammatory activities. It has been shown that oridonin can also trigger FBXW7, increasing expression levels and resulting in c-Myc degradation, consequent induction of apoptosis and growth inhibition of the K562 cells-derived xenograft model. However, despite its attractive pharmacological profile of safety and efficacy, oridonin has not been widely adopted into clinical practice due to its relatively moderate potency, imprecise mechanisms of action, limited aqueous solubility and oral bioavailability. Nevertheless, oridonin derivatives showing higher potency and better pharmacological properties have been synthesized and are currently being testing in clinical trials for the treatment of leukemia [97].

### 5.2. Inhibiting the Aminopeptidases

Aminopeptidases are a class of metalloenzymes (usually zinc) implicated in the final step of intracellular protein degradation by hydrolyzing *N*–terminal amino acids of peptides, generated by the UPS, into free amino acids, which can be recycled for renewed protein synthesis [98]. Aminopeptidases are essential for several physiologically important processes, including protein maturation, degradation of peptides and cell cycle control. Additionally, due to their pleiotropic nature, aminopeptidases have been found implicated in several pathologies, including cancer [99]. In fact, for cancer cells, the supply of cellular free amino acids, regulated by aminopeptidases, is of primary importance for their survival and proliferation. Furthermore, cancer cells can be also dependent on specific amino acids and depletion of them has a greater impact on cancer cells survival than normal cells [100]. For example, it has been showed that proliferation of AML and MM cells is strongly dependent on the unfolded protein response regulated by aminopeptidases. Accordingly, aminopeptidase inhibition results in a marked suppression of MM and AML cell growth and survival [101]. The critical relevance of aminopeptidase for cancer progression paved the way for investigating inhibitors of aminopeptidases as potential antileukemic therapeutic drugs.

The critical relevance of aminopeptidase for cancer progression paved the way for investigating inhibitors of aminopeptidases as potential antileukemic therapeutic drugs.

Aminopeptidase inhibitors exert their effects by inducing the amino acid deprivation response (AADR), which is involved in transcriptional and post-transcriptional regulatory mechanisms, including overexpression of amino acid synthetic genes, amino acid transporters and tRNA synthetases, and by inhibiting mTOR proliferative pathway [102,103].

The first clinically approved aminopeptidase inhibitor, named bestatin, was thought of as an immunomodulating drug; however, later on, it has been demonstrated that bestatin also exhibited antiproliferative effects in several cancer cell lines, including leukemic cells, highlighting its activity as an anticancer drug [104]. Tosedostat (CHR2797) is a hydrophobic aminopeptidase inhibitor prodrug that is rapidly absorbed by cells and then activated into a hydrophilic pharmacologically active product. It is able to inhibit multiple aminopeptidases, with preference for leukotriene A-4 hydrolase, aminopeptidase N and leucine aminopeptidase [105]. Tosedostat exerts its anticancer effects by depleting intracellular amino acids and triggering AADR, finally resulting in the suppression of cell growth [105]. Tosedostat has showed marked in vitro cytotoxicity in AML samples, compared to normal bone marrow progenitors [106]. Tosedostat has been tested in clinical trials for the treatment of AML in combination with chemotherapeutic drugs and has shown promising results (Table 1) [107]. Recently, it was found that statins can potentiate the antitumor activity of tosedostat in AML. Mechanistically, it has been found that the dual inhibitory effect of impaired Ras homolog enriched in brain (RHEB) prenylation by statins and CHR2863-induced mTOR inhibition achieves a potent synergistic inhibition on human AML cells [108].

### 5.3. Inhibiting the Proteasome

Proteasome inhibition has become a clinically validated strategy for the treatments of MM and mantle cell lymphoma (MCL) [109].

Bortezomib was the first PI to enter in clinical trials and the first PI to be brought into clinical use for the treatment of MM and MCL. Bortezomib is a reversible inhibitor of β5 proteasomal proteases. Mechanistically, bortezomib induces the stabilization of IκB, increasing protein levels of proapoptotic factors and accumulation of negative cell cycle regulators including p53 and p27 [110,111,112]. Since its FDA (The Food and Drug Administration) approval, the efficacy of bortezomib has been tested in combination with other drugs, showing promising results in the relapsed and/or refractory myeloma settings and in AML treatment [113]. For example, bortezomib co-treatment alongside chemotherapeutic drugs such as doxorubicin is currently being evaluated in clinical trials for AML (Table 1) [114].

To improve the therapeutic index relative to bortezomib, carfilzomib, an irreversible β5 inhibitor, was developed, giving rise to the possibility of long-duration proteasome inhibition. Carfilzomib was approved in 2012 by the FDA as a single agent for the treatment of MM in patients who have received at least two prior lines of therapy, and with disease that are refractory to the most recent line of treatment [115].

A second generation of PIs was developed with the main goal of improving their oral bioavailability compared to the first generation (bortezomib and carfilzomib). Among these oral PIs, ixazomib is the most advanced in clinical development, being FDA approved in 2015. It is an orally available analog of bortezomib, therefore allowing frequent dosing for maintaining a constant level of proteasome inhibition. It induces apoptosis in MM cells by enhancing the expression of proapoptotic genes [116]. Oprozomib (ONX0912; PR-047) was designed as a tripeptide analog of carfilzomib. In contrast to intravenous administration of bortezomib and carfilzomib, oprozomib retained a better oral bioavailability, becoming suitable for oral administration. It showed antitumor activity, potency and selectivity similar to carfilzomib in MM and can be used to treat bortezomib-, dexamethasone- or lenalidomide-resistant MM. Additional preclinical and clinical studies of these agents are currently ongoing in several settings, including AML and ALL with promising results [117].

### 5.4. Clinical Trials Exploring Proteasome Inhibitors in Leukemic Patients

Following FDA-approved indications for MM and MCL treatment, the effects of PIs, especially bortezomib, have been studied in several clinical trials for different hematological malignancies, either as a single agent or in combination with other drugs (Table 1). However, despite the promising results in MM and in preclinical models of leukemia, bortezomib showed only a modest clinical activity as a single agent in AML trials, while a better outcome resulted from the combination therapies. For example, phase I and II studies exploring the combination treatment of bortezomib and idarubicin plus cytarabine highlighted that 65% of patients achieved complete remission and an additional 4% achieved partial remission. Additionally, a significant neuropathy was reported in 12% of the patients [118,119]. A controversial clinical efficacy, as well as adverse side effects linked to combination therapy with bortezomib, have also emerged in leukemia, including pediatric leukemia. A phase I study, in which bortezomib was combined with conventional chemotherapeutic drugs, showed promising clinical activity in adult AML as well as pediatric ALL patients. Conversely, a phase II study where bortezomib was combined with re-induction chemotherapy in relapsed pediatric AML patients did not show improvement of the overall survival [120].

Additionally, despite pre-clinical efficacy, a phase II trial (ClinicalTrials.gov Identifier: NCT00818649) investigating the combination of bortezomib with histone deacetylase inhibitor SAHA was ended due to high toxicity. Altogether, the clinical use of PIs in leukemia appears to offer favorable results, especially when combined with chemotherapy regimens, although disease relapses remain a significant impediment to the complete remission of both acute adult and pediatric leukemia. Additionally, side effects such as infections and neutropenia were commonly found as adverse drug reactions, therefore limiting clinical efficiency of PIs. The development of the second-generation PIs carfilzomib, displaying not only an in vitro antileukemic activity consistently higher than bortezomib but also a low rate of treatment-associated neuropathy, opened an exciting perspective for its clinical evaluation study. Carfilzomib is currently undergoing testing in phase I clinical trials as a potential PI for treatment of leukemia (ClinicalTrials.gov Identifier: NCT01137747) [121]. For an extensive review focused on clinical trials investigating PIs in leukemia, refer to Niewert et al. [122], Csizmar et al. [109] and Cloos et al. [117].

### 5.5. Inhibiting DUBs

DUB deregulation contributes to cancer; thus, several DUB inhibitors displaying anticancer activity have been synthesized so far, although none of them have reached the stage of clinical experimentation.

Inhibition of USP7 has been shown to activate apoptotic pathways in tumor cells [123]. HBX41108 and its derivatives HBX19818 were shown to covalently bind to the catalytic Cys residue of USP7 and to stabilize p53, promoting G1 arrest and consequent apoptosis of cancer cells. Later on, compounds P5091 and P22077 were tested as specific USP7 inhibitors in in vitro and in vivo settings. P5091 was found to induce apoptosis in various MM cell lines as well as ex vivo cells, including those resistant to previous treatments [124].

Re-evaluation of HBX19818 and P22077 revealed that these molecules can also potently inhibit USP10 deubiquitinase. Flt3 degradation mediated by HBX19818 and P22077 showed high efficiency in treatment of AML cell lines and patient-derived AML xenograft resistant to FLT3 kinase inhibitors, highlighting USP10 inhibition as a potential target for Flt3 kinase inhibitor-resistant AML patients [73].

Among the compounds targeting USP9X, a non-specific inhibitor of USP9X, named (EOAI3402143) G9, belonging to the USP9X-second generation inhibitors, has been shown to destabilize the pro-survival protein MCL1 and increased p53 levels, promoting apoptosis in a dose-dependent manner and reducing tumor growth of human myeloma xenograft [125]. However, despite these promising results, further studies are still needed to translate the G9 compound into clinical trials.

Proteasomal DUBs have been correlated with cancer progression by affecting cell proliferation as well as DNA damage response. b-AP15 and its derivative VLX1570 were found as inhibitors of ubiquitin-specific peptidase 14 (USP14) and, to a lesser extent, UCHL5. VLX1570 treatment was able to accumulate ubiquitinated substrates and exhibited excellent efficacy in in vivo and in vitro AML models as well as MM models [126,127]. Nevertheless, clinical trial evaluating VLX1570 and dexamethasone efficacy for the treatment of MM has currently been suspended in phase 1/2 due to dose-limiting toxicity (ClinicalTrials.gov Identifier: NCT02372240) (Table 1).

### 5.6. Enhancing UPS

Genomics advances have identified an increasing number of proteins implicated in cancer that may also be used as potential therapeutic targets. Despite the recent pharmacological progresses, many of the most attractive oncoproteins, such as c-Myc, are still far from being considered chemically druggable. However, new insights in the UPS have identified many undruggable oncoproteins as molecular targets of specific E3 Ub-ligase; therefore, enhancing the ubiquitination and subsequent degradation of these targets has emerged as promising therapeutic avenue for cancer therapy.

### 5.7. Activating E3 Ubiquitin Ligases: The Immunomodulatory Drugs

Thalidomide and its analogs, lenalidomide and pomalidomide, are collectively known as immunomodulatory or ‘‘IMiD’’ drugs. IMiDs can bind to the E3 ligase cereblon (CRBN) 5, forming a complex promoting the ligand-dependent degradation of neo-substrates, including Ikaros family zinc finger proteins 1 and 3 (IKZF1, IKZF3) and casein kinase-1a (CK1α). IKZF1 and IKZF3 are two transcription factors that play a critical role in MM progression [128], and it is not surprising that CRBN modulation via IMiBs is being validated as a tool for the treatment of MM. In fact, the therapeutic combination of bortezomib and IMiDs confers a favorable prognosis in MM [129]. Unfortunately, emerging evidences have showed that cancer cells may acquire resistance to the MM IMiD drug lenalidomide upon mutations of a single amino acid in IKFZ3 that rescues this transcription factor from proteolytic degradation [130]. Avadomide (also known as compound CC-122) is an IMiD derivative recently synthesized. CC-122 is currently being tested in phase 1/2 clinical trials for the treatments of hematological malignancies and solid tumors (Table 1). It shows a better cellular potency in inducing ubiquitination and consequent proteasomal degradation of IKZF1 and 3 compared with previous IMiDs [131].

SPLAM (SPLicing inhibitor sulfonAMides) molecules, including indisulam, CQS and tasisulam, share an internal sulfonamide as a common structure motif. Similar to IMiDs, it was recently found that SPLAMs can bind and stabilize the interaction between the E3 ligase CRL4-DCAF15 and the cancer-associated neo-substrates RNA-binding motif protein (RBM) 39 as well as its closely related splicing factor RBM23 leading with their polyubiquitination and consequent proteasomal degradation. Degradation of RBM39 or RBM23 led to aberrant pre-mRNA splicing, including intron retention and exon skipping in hundreds of genes [132,133]. SPLAMs did not advance in clinical test since fewer than 15% of patients had a clinical response, and the objective response rates were less than 40% in solid tumors; however, clinical trials are currently focusing on leukemia and lymphoma since these diseases express high levels of DCAF15 protein.

### 5.8. Activating E3 Ubiquitin Ligases: Protein-Targeting Chimeric Molecules

The protein-targeting chimeric molecules (PROTACs) concept is based on generating artificial molecules to recruit a specific E3 Ub-ligase to a determined target protein. They function as heterobifunctional molecules connecting an E3 ligand to a substrate protein ligand with an optimal linker (Figure 2) [134]. Therefore, PROTACs promote the formation of the ternary complex comprising the protein of interest (POI), the E3 and the PROTAC itself, resulting in ubiquitination and proteasomal degradation of the POI [135]. Proof of concept studies led to the creation of PROTAC-1 molecule that is formed by a ten amino acids phosphopeptide degron from IkBα, which is recognized by E3 SCFβ-TRCP ligase and covalently linked with ovalicin, a small molecule inhibitor of methionine aminopeptidase-2 (MetAP2). Strikingly, it was showed that in the presence of PROTAC-1, MetAP-2 is recruited to SCFβ-TRCP, with its consequent ubiquitination and degradation in a PROTAC-1-dependent manner [136].

Despite their success, the first generation of PROTACs suffered from low potency, mainly due to their poor permeability and high protease susceptibly commonly associated with peptide-based therapeutics. However, the conception of specific ligands for E3 ligases has highly improved this technology, providing more drug-like molecules. The second generation of PROTAC is currently a very active research field, and many compounds have been synthesized so far; thus, it is impossible to cover all efforts for describing all PROTAC compounds. In this review, we will focus on some most promising examples listed below applied to leukemia therapy.

Tyrosine kinase inhibitors (TKIs) are currently used to treat BCR-ABL-driven chronic myeloid leukemia (CML). However, point mutations in the tyrosine kinase domain of BCR-ABL have often occurred with consequent drug resistance insurgence. For creating BCR-ABL degrader compounds, researchers conjugated BCR-ABL TKIs dasatinib, which binds the c-ABL kinase domain to either the von Hippel–Lindau (VHL) or CRBN E3 ubiquitin ligase ligands. The resulting bifunctional compound was able to bind BCR-ABL via the TKI moiety and either VHL or CRBN via its recruiting ligand [137]. Subsequently, SIAIS178, also another BCR-ABL degrader, disclosed a comparable activity. Both compounds have displayed cell growth arrest and proliferation inhibition of BCR-ABL-driven leukemia [138].

PROTAC technology has also been applied for degrading the oncogenic protein bromodomain-containing protein 4 (BRD4). BRD4 has been reported as a required target for leukemia progression by promoting a cancer-associated transcriptional program [139]. PROTAC dBET1 was conceived using a phthalimide ring as CRBN E3 ligase recruiter linked with the BRD4 inhibitor JQ1. dBET1 treatment induces strong BRD4 degradation, resulting in apoptosis of leukemia cell lines as well as ex vivo leukemic cells [140]. Similarly, ARV-825, another PROTAC compound, by recruiting CRBN also aimed to degrade BRD4. This compound showed a better efficacy in deregulating c-Myc compared to dBET1, resulting in a more effective cell proliferation inhibition and apoptosis induction [141].

Bromodomain-containing protein 9 (BRD9), a component of SWItch/Sucrose Non-Fermentable (SWI/SNF) remodeling complex, was found implicated in leukemia development and progression by regulating expression of cancer-related genes; thus, targeting BRD9 is considered a promising therapeutic strategy for leukemia [142]. Given that, PROTAC targeting BRD9 (dBRD9) was developed as bifunctional degrader by conjugating a BRD9 chemical ligand with CRBN E3 ligase ligand. In vitro studies showed that dBRD9 induced a rapid BRD9 degradation via proteasome and exerted a potent antiproliferative effect in AML cell lines. Later on, a new BRD9 bifunctional degrader named VZ185 was developed. VZ185 showed a stronger anticancer activity in leukemic cell lines at very low concentration compared to dBRD9 [143].

Signal transducer and activator of transcription 3 (STAT3) is constitutively activated in different human cancers and it regulates a set of genes implicated in cancer cell survival, proliferation, metastasis and drug resistance, hence, it is a powerful cancer therapeutic target. Although STAT3 inhibitors have reached the clinical development stage, they demonstrated very limited clinical activity [144]. Recently, researchers, by tethering a SI-109, a chemical binder of STAT3, with a ligand for either CRBN or VHL, synthesized a potent and specific PROTAC compound targeting STAT3 named SD-36. SD-36 selectively degrades STAT3 over other STAT proteins and exerts anticancer activity in preclinical models of acute myeloid leukemia and anaplastic large cell lymphoma. Particularly, the SD-36-mediated STAT3 degradation induced in a strong suppression of its transcription network, resulting in cell cycle arrest and apoptosis induction in leukemia and lymphoma cells [145].

Although PROTACs is a promising technology for drug discovery, especially for targeting the “undruggable” proteome, it faces important challenges for the future. For example, it is essential for identifying suitable ligands to successful design PROTACs, particularly for E3 Ub-ligase–oncoprotein interactions. Furthermore, the pharmacokinetic and pharmacodynamic characteristics of PROTACs are not very clear in clinical practice; it is likely that they require further improvement. Additionally, another possible challenge for PROTAC development is the drug resistance in treated cells due to genomic alterations of the core component of E3 ligase complexes [146].

## 6. Conclusions

Protein stability deregulation of oncogenes and/or oncosuppressors is critical for cancer development and progression. Thus, our understanding of the ubiquitin code may contribute to the development of promising therapeutic agents. To date, there is enormous therapeutic potential in targeting UPS system. The exciting results reached with the use of the PIs, in some types of blood cancers, have contributed to the development of additional drugs targeting other components of UPS, although targeting UPS effectors suffers of target specificity since they can affect thousands of proteins. Furthermore, UPS components such as E3 enzymes lack a well-defined activity pocket; thus, the development of specific inhibitors remains challenging even though new advances in drug discovery, as well as development of new screening technologies, are surprisingly improving our abilities to drug demanding targets.

Controlling the function of a certain protein by controlling its protein levels represents a promising new therapeutic approach in modern drug discovery. Enhancing protein degraders can extend horizons of druggable targets to those proteins that are still considered undruggable and are currently limiting medicinal research. Therefore, it will be interesting to see whether attractive oncogenic protein drivers such as c-Myc can eventually become part of a druggable proteome. PROTAC offers an excellent opportunity to target and degrade many difficult oncoproteins, suggesting that PROTAC could be developed as a new technology platform for drug discovery. The major challenge will be identifying suitable ligands for binding undruggable proteins as well as optimizing the linker molecules that connect the two chemical (PROTAC) moieties.

Targeting UPS by either inhibiting or enhancing UPS effectors has proved to be one promising approach for the treatments of hematological malignancies. However, a better understanding of the mechanisms underlying E3 regulation and function in leukemia and an increase efficacy of drug discovery technologies are expected to open and expand the generation of new blood tumor therapies.

## Figures and Tables

**Figure 1 cancers-12-01898-f001:**
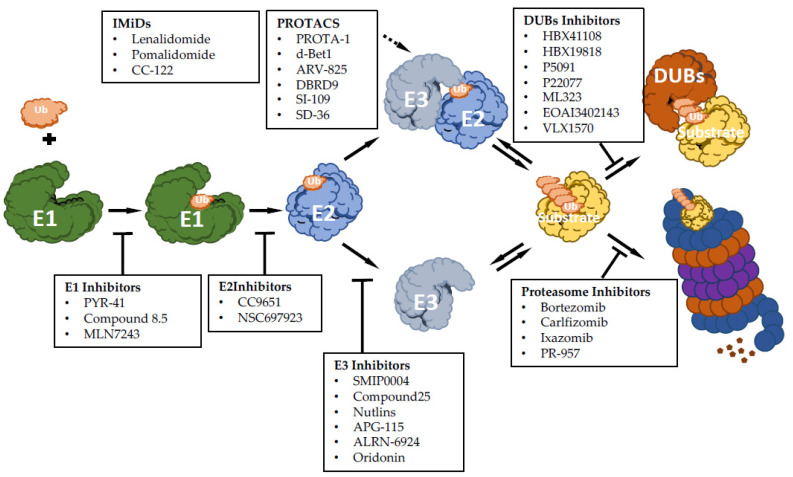
Summary scheme of discussed drugs and their function: E1 enzyme inhibitory molecules, E2 enzyme inhibitory molecules, E3 enzyme inhibitory molecules, immunomodulatory drugs of E3 enzymes (IMiDs), proteolysis targeting chimeras molecules (PROTACs), proteasome inhibitor drugs and deubiquitinase inhibitor molecules (DUBs).

**Figure 2 cancers-12-01898-f002:**
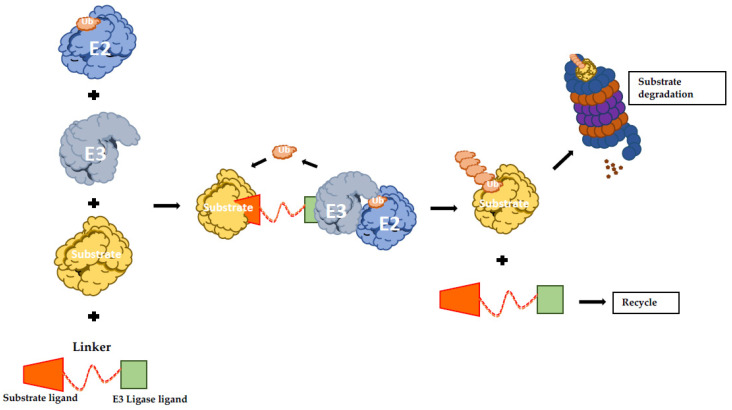
Schematic representation of PROTAC-induced degradation. The bifunctional hybrid molecule binds to both E3 ubiquitin ligase and the target protein. Following E3-mediated ubiquitination, the substrate undergoes UPS-mediated protein degradation. UPS: ubiquitin proteasome system; PROTAC: protein-targeting chimeric molecules.

**Table 1 cancers-12-01898-t001:** Reported E1, E3, proteasome, deubiquitinase enzyme and aminopeptidase modulators in clinical trials.

Modulators	Target	Clinical Stage
E1 enzyme modulators		
MLN7243	Ubiquitin-Activating Enzyme (Uba1)	Phase 1 completed
E3 enzyme modulators		
APG-115	MDM2-p53	Phase 1 recruiting
ALRN 6924	MDM2\MDMX	Phase 1
CC-122	Cereblon (CRBN)	Phase 1\2
Indisulam	Cullin-4A-DCAF15	Phase 2 completed
CQS		
Tasisulam		Phase 3 terminated
Proteasome Inhibitors alone or in combination with other drugs		
Bortezomib + Idarubicin + Cytarabine	Proteasome B5 protease	Phase 1
Bortezomib + Daunorubicin+ Cytarabine		
Bortezomib + Decytabine		Phase 2
Bortezomib + SAHA		Phase 2 suspended
Carlfizomib		Phase 1
Deubiquitinase inhibitors modulators		
VLX1570	Ubiquitin Specific Peptidase 14,Ubiquitin C-terminal hydrolase 37	Phase 1\2 suspended
Aminopeptidase inhibitors in combination with other drugs		
Tosedostat + Decytabine	Leukotriene A4LTA4, Aminopeptidase N, Leucine aminopeptidase PuSA	Phase 2
Tosedostat + Cytarabine		
Immunomodulatory (IMiDs) drugs		
CC-122	Ikaros family zinc finger protein 1 IKZF1 and IKZF3	Phase 1\2

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
