# Peer review of "The Ubiquitin Proteasome System in Hematological Malignancies: New Insight into Its Functional Role and Therapeutic Options"

_cancers, 2020, doi:10.3390/cancers12071898_

Round 1

Reviewer 1 Report

The review written by Antonella Di Contanzo et al explores the role of the UPS in hematological malignancies as well as discussing therapeutic approaches targeting the UPS for therapy. It is clearly written and flows nicely. I enjoyed reading section 5.5 dealing with SPLAM as many review do not hit on these topics.

One thing that I do think would be helpful to include is a section on the UPS system in relation to virus induced cancers.  The review does not discuss cancers due to infection.  HBV, HTLV, EBV, and KSHV come to mind and all modulate the UPS system and result in hematological malignancies.  EBV and KSHV encode their own DUB and have been shown to involved in evasion or suppression of the immune system as well as targeting DNA repair pathways, including some of the same proteins mentioned in section 3.1 and 4.

Reviewer 2 Report

In the review by Costanzo et. al. they review the current literature of the ubiquitin proteasome system in hematopoietic malignancies. The authors focused on both ubiquitin ligases and deubiquitinating enzymes, along with current therapeutics. Although the topic is very interesting due to the increased knowledge, literature, and current therapeutics targeting of both the proteasome system and ubiquitin E3 ligases the review contains major flaws. I recommend revision prior to acceptance. A number of issues include;

  • The review contains a large number of topics, and would benefit from more focus (ie; focus on either functional role or therapeutics). Due to the broad range the authors were unable to really go into details. As an example, the authors do not expand on FBXW7 in hematopoietic malignancies. In addition they do not discuss other known ligases or DUBS like KLHL6 in lymphoma.
  • The authors did not present a complete review of clinical trial with proteasome inhibitors. The use of proteasome inhibitors in AML have mixed results in trials, and in the case of pediatric AML, the results have been negative and major side affects have been seen.
  • In the UPS machinery section mono and poly ubiquitination are lumped together. The authors need to separate out proteasomal degradation from ubiquitination and split this into two sections.
  • Attention needs to be paid to nomenclature, when discussing genes and proteins. These need to be consistent and accurate.

Reviewer 3 Report

The authors give a quite comprehensive overview of how the UPS is involved in the development of hematological malignancies and the potential of interfering in the UPS to affect malignant cells.

The manuscript is very detailed on many facets of the UPS and thereby an helpful asset for interested researchers.

I have some comments:

  • To complete the story of protein degradation, the authors may consider to include aminopeptidase inhibitors (acting after proteasomal degradation) that also show promise in killing leukemia cells.
  • There are several typographical errors that need to be re-checked.
  • In Figure 1, I would avoid abbreviations and if not possible, give the clarification in the legend to be able to understand the figure without reading the complete text. (a space is missing after E2 in the figure). The legend text of both figures is actually very limited and the authors might consider to elaborate, so the figure can be understood as separate entity.
  • In lines 260 and 547 the remarks would benefit from a specific reference.
  • First the gene NFKBIA was mentioned, while later on IkB was mentioned. Is there a reason to choose either of them? Perhaps good to first mention them both and use one to further address to in the text.
  • Perhaps the paragraph of 5.2.1 was based on the request of another reviewer (text in red), but since there is no 5.2.2 etc it seems quite odd to have this text in a separate paragraph.

Author Response

Response to Reviewer 3 Comments

Point 1: To complete the story of protein degradation, the authors may consider to include aminopeptidase inhibitors (acting after proteasomal degradation) that also show promise in killing leukemia cells.

Response 1: We thank the reviewer for this interesting suggestion.

This is an interesting topic, so we added to the text a section describing the relevance of aminopeptidase in cancer progression and some of the aminopeptidases inhibitors developed as potential anti-leukemic therapeutic drugs.

Point 2: There are several typographical errors that need to be re-checked.

Response 2: We thank the reviewer for pointing out these mistakes. The text has been corrected.

Point 3: In Figure 1, I would avoid abbreviations and if not possible, give the clarification in the legend to be able to understand the figure without reading the complete text. (a space is missing after E2 in the figure). The legend text of both figures is actually very limited and the authors might consider to elaborate, so the figure can be understood as separate entity.

Response 3: Again, we thank the reviewer for this suggestion. We revised again the figure legends text, trying to provide more insight to making them more understandable

.

Point 4:  In lines 260 and 547 the remarks would benefit from a specific reference.

Response 4: We added to the text the specific reference.

Point 5:  First the gene NFKBIA was mentioned, while later on IkB was mentioned. Is there a reason to choose either of them? Perhaps good to first mention them both and use one to further address to in the text.

Response 5: Thanks the reviewer for the observation. We used two different nomenclature: NFKBIA to indicate the gene, where we referred to gene, and IkBa (the related protein) when we referred to the protein. To be more clear, we mentioned them only at first time and, subsequently, we used IkBa.

Point 6: Perhaps the paragraph of 5.2.1 was based on the request of another reviewer (text in red), but since there is no 5.2.2 etc it seems quite odd to have this text in a separate paragraph.

Response 6: Accordingly with reviewer, we revised the text naming the clinical trial subsection as separate paragraph.

Reviewer 4 Report

In gerneral, this is quite a comprehensive review of the UPS in relation to haematological malignancies. As this is such a broad topic that has been covered to different extents in previous reviews, a bit of restructuring may be of benefit this review to distinguish it from previous reviews on the topic. For example, as mentioned in the review, it is impossible to try and cover all E3 ligases displaying aberrant expression in blood cancers and likewise DUBs, therefore rather than trying to superficially go into defining a role for aberrant E3s/DUBs and then duplicating some of this information on the section on inhibiting these enzymes, perhaps these could be condensed into one section. Where the review demonstrates most novelty is in detailing the emerging field of PROTACS and this is perhaps worth a bit more emphasis. 

Inclusion of a table would be useful to collate information of compounds targeting the UPS that are approved or in clinical trials.

Be consistent with use of abbreviations, for example, when referring to lysine linkages sometimes 'K' is used to denote lysine and sometimes 'lys' is used. 

Ensure the correct referencing style is used, there are some instances (for example on line 314 and 400) where a web link is used instead of or in addition to a numerical reference.

Author Response

Response to Reviewer 4 Comments

Point 1: In gerneral, this is quite a comprehensive review of the UPS in relation to haematological malignancies. As this is such a broad topic that has been covered to different extents in previous reviews, a bit of restructuring may be of benefit this review to distinguish it from previous reviews on the topic. For example, as mentioned in the review, it is impossible to try and cover all E3 ligases displaying aberrant expression in blood cancers and likewise DUBs, therefore rather than trying to superficially go into defining a role for aberrant E3s/DUBs and then duplicating some of this information on the section on inhibiting these enzymes, perhaps these could be condensed into one section.

Response 1:We thank the reviewer for this useful comment.

Following a long reflection ,we decided to keep a schematic description of UPS (molecular aspects distinct from therapeutic approaches) because we strongly believe that this would help the reader to orient himself along the variety of topics. However as suggested, we revised the “targeting UPS” section by removing redundant information already reported previously in the review to make the reading non-repetitive.

Point 2: Where the review demonstrates most novelty is in detailing the emerging field of PROTACS and this is perhaps worth a bit more emphasis

Response 2:We thank the reviewer for this comment.

How highlighted from the reviewer, this study space from describing molecular bases of UPS pathway to explore the different classes of UPS targeting drugs. Thus, given the vastness and the number of topics addressed, and also considering the limited number of words, we tried to provide an overview of the most important topics concerning the UPS in hematological malignancies. For these reasons we conceptualized PROTAC compounds focusing essentially on leukemia and mainly describing the molecular basis of their mechanisms of action.

Point 3: Inclusion of a table would be useful to collate information of compounds targeting the UPS that are approved or in clinical trials.

Response 3: Thanks to reviewer for this suggestion. We included a table which schematizes the different molecules targeting the UPS components and aminopeptidases in clinical trial (Table 1).

Point 4: Be consistent with use of abbreviations, for example, when referring to lysine linkages sometimes 'K' is used to denote lysine and sometimes 'lys' is used.

Response 4: We thank the reviewer for pointing out these mistakes. The text has been corrected.

Point 5: Ensure the correct referencing style is used, there are some instances (for example on line 314 and 400) where a web link is used instead of or in addition to a numerical reference.

Response 5: We added to the text the specific references.

Round 2

Reviewer 2 Report

Point 1: The review contains a large number of topics, and would benefit from more focus (ie; focus on either functional role or therapeutics). Due to the broad range the authors were unable to really go into details. As an example, the authors do not expand on FBXW7 in hematopoietic malignancies. In addition they do not discuss other known ligases or DUBS like KLHL6 in lymphoma.

Although the authors discuss both FBXW7 and KLHL6, no additional ligases or DUBS are included. 

Author Response

Thanks